# Diagnosis and Treatment of Pulmonary Disease in Sea Turtles (*Caretta caretta*)

**DOI:** 10.3390/ani10081355

**Published:** 2020-08-05

**Authors:** Stefano Ciccarelli, Carmela Valastro, Antonio Di Bello, Serena Paci, Francesco Caprio, Maria Laura Corrente, Adriana Trotta, Delia Franchini

**Affiliations:** Department of Veterinary Medicine, University of Bari “Aldo Moro”, 370010 Valenzano, Italy; stefano.ciccarelli@uniba.it (S.C.); carmela.valastro@uniba.it (C.V.); serena.paci@uniba.it (S.P.); francesco.caprio@uniba.it (F.C.); marialaura.corrente@uniba.it (M.L.C.); adriana.trotta@uniba.it (A.T.); delia.franchini@uniba.it (D.F.)

**Keywords:** antibiotic resistance, BAL (bronchoalveolar lavage), endoscopy, pulmonary disease, sea turtle

## Abstract

**Simple Summary:**

Evaluation of the respiratory system is a critical aspect of sea turtle medicine due to their peculiar anatomy. The location of the lungs under the carapace makes them vulnerable to wounds involving the carapace, which are frequent and significant complications of vessel strike injuries. Open lung wounds result in aspiration, loss of buoyancy control, and secondary infection. In sea turtles, pulmonary diseases can originate from many causes. Increased marine pollution is reflected in the increasing occurrence of sea turtles with entanglement injuries around the neck or flippers, caused by plastic waste, ghost nets, fishing lines, etc. These injuries may directly damage the upper airway or create life-threatening secondary infections. The object of the study was to describe clinical signs, radiographic, endoscopic and computed tomography examinations, and cytological and microbiological findings useful to obtaining an accurate diagnosis of pulmonary diseases in sea turtles. Moreover, we describe the treatment carried out on the basis of antimicrobial susceptibility testing to avoid unnecessary treatments and antibiotic-resistance phenomenon. Data were collected from 14 turtles. Radiographic signs of pulmonary pathology were seen in all cases. Four sea turtles underwent advanced diagnostic investigations so as to better characterize the pulmonary pattern. The bronchoalveolar lavage allowed us to withdraw fluid from the lower airways and to perform cytological and bacteriological examinations on all 14 subjects. The study suggests how it may be useful to implement diagnostic procedures in order to obtain an accurate and early diagnosis, to prevent unnecessary therapy, and to contain antibiotic-resistance phenomena.

**Abstract:**

The aim of this study was to describe the clinical signs, radiographic, endoscopic and CT findings, cytological and microbiological findings and treatments of pulmonary diseases in sea turtles, in order to obtain an accurate diagnosis that avoids unnecessary therapy and antibiotic-resistance phenomena. In total, 14 loggerheads (*Caretta caretta*), with clinical and/or radiographic findings of pulmonary pathology, were assessed through various combinations of clinical, radiological, CT, endoscopic examination and bronchoalveolar lavage, which recovered fluid for cytologic and microbiologic analysis. In all cases, radiographic examination led to a diagnosis of pulmonary disorders—4 unilateral and 10 bilateral. All bacteria cultured were identified as Gram-negative. Antibiotic resistance was greater than 70% for all beta-lactams tested. In addition, all bacterial strains were 100% resistant to colistin sulfate and tetracycline. Specific antibiotic therapies were formulated for seven sea turtles using Enrofloxacin, and for four sea turtles using ceftazidime. In two turtles, antibiotic therapy was not included due to the presence of antibiotic resistance against all the antibiotics evaluated. In both cases, the coupage technique and environmental management allowed the resolution of the lung disease without antibiotics. All 14 sea turtles were released back into the sea. Radiographic examination must be considered the gold standard for screening sea turtles that show respiratory signs or abnormal buoyancy. Susceptibility testing with antimicrobials allowed appropriate therapy, including the reduction of antibiotic-resistance.

## 1. Introduction

The health and welfare of sea turtles is a sensitive topic, due to their increased exposure to human activities, both directly and indirectly. Respiratory diseases are an important and common pathology. Sea turtles manifest respiratory disease differently than mammals; not through the characteristic clinical signs of dyspnea, cyanosis of the mucous membranes, nasal discharge and cough, but mainly through buoyancy disorder. In some cases, sea turtles are asymptomatic in the first clinical evaluation. In sea turtles, pulmonary diseases can originate from many causes. Foreign bodies, such as fishhooks, may easily penetrate the trachea and bronchi, entering through the mouth or the esophagus. Increased marine pollution is reflected in the increasing occurrence of sea turtles with entanglement injuries around the neck or flippers caused by plastic waste, ghost nets, fishing lines, etc. These injuries may directly damage the upper airway or create life-threatening secondary infections. Turtles with flipper impairments have difficulty swimming, diving and surfacing to breathe [1]. The location of the lungs within the dorsal coelomic cavity makes them vulnerable to wounds involving the carapace, which are frequent and significant complications of vessel strike injuries. Open lung wounds result in aspiration, loss of buoyancy control and secondary infection [2].

Over the last few decades, some authors have reached the conclusion that the inhalation of water can be considered as one of the primary causes of bacterial and fungal pneumonia in sea turtles. Moreover, any debilitating condition can lead to secondary bacterial, fungal, viral (i.e., fibropapilloma) or parasitic pneumonia [3,4]. For this reason, a correct diagnosis of lung disease as early as possible is essential. Based on this background information, evaluation of the respiratory system is a critical aspect of sea turtle medicine. Due to the anatomy of these animals, auscultation is not reliable for the detection of respiratory problems in sea turtles, whereas listening to the character of inspiration and expiration can be informative. During clinical observation of the respiratory system in healthy turtles, consistent crackling or gurgling should not be heard. 

When feasible, the physical exam should include an in-water evaluation to assess symmetry of buoyancy, as well as swimming and diving ability. Abnormal buoyancy can result from pulmonary disease, as well as neurological disease, pneumocoelom, or gas and/or foreign bodies within the gastrointestinal tract.

Bronchoalveolar lavage (BAL) is a sampling technique that is currently considered to be a fundamental collateral examination in the evaluation of respiratory diseases. The technique literally means to wash, through irrigation and subsequent re-aspiration of sterile saline solution (NaCl 0.9%), the lower airways, in order to recover cells and exudates, which will be subsequently subjected to cytological and microbiological examination. BAL is ideally performed bronchoscopically, but can be performed “blindly” through a sterile catheter or bronchoscopically, depending on the instrumentation available, and the samples obtained are typically representative of upper airways. 

In the last two years, the literature in the field of sea turtles has provided indications for sampling techniques, methods for sample storage and preparation, and the interpretation of the cytological framework [5].

The purpose of the study was to describe clinical signs, findings from radiographic, endoscopic and CT examinations, and cytological and microbiological findings that are useful to obtaining an accurate diagnosis of pulmonary diseases in sea turtles. We also describe the treatment carried out on the basis of antimicrobial susceptibility testing, which avoids unnecessary treatments and antibiotic-resistance phenomenon.

## 2. Materials and Methods

Between October 2016 and June 2019, 14 loggerhead sea turtles (*Caretta caretta*) were referred to the Sea Turtle Clinic (STC) of the Department of Veterinary Medicine of the University of Bari “Aldo Moro”, after having been taken to local Adriatic and Ionian Sea turtle rescue centers, for buoyancy issues, carapace lesions and flipper and/or neck entanglements. The animals were found drifting at sea, stranded on the coast or were incidentally captured (bycatch). 

Upon admission, physical examination of each turtle was performed, including morphometrics (curve carapace length (CCL) notch to tip, curve carapace width (CCW) and weight) as well as measurements of core body temperature from the cloaca and blood sampling to measure the plasma biochemical and hematological values. The turtles were kept in tanks with saltwater at 36‰ (salinity percentage of the Adriatic Sea) in a room at approximately 25–30 °C until clinical improvement.

All loggerhead sea turtles underwent dorso-ventral (DV), latero-lateral (LL) for each side, and craniocaudal (CrCd) radiographic examinations. To better visualize the pulmonary parenchyma, a total-body multi detector computed tomography (MDCT) was performed on four sea turtles, with a 16-slice MDCT scanner (Somaton Emotion, Siemens, Forchheim Germany). The technical scan and reconstruction parameters were 110 KVp, 180 mAs, 1-mm slice thickness, pitch of 0.8, 0.6 s/rotation, 0.5 mm reconstruction interval, and standard (pulmonary) acquisition algorithm. Three-dimensional (3D) multi-planar reformatted, maximum-intensity projection, and volume-rendered images were obtained using a dedicated 3D software program (Pixmeo OsiriX DICOM viewer^®^, Pixmeo, Bernex, Switzerland). All turtles received general anesthesia, with 5 mg/kg propofol (Propovet^®^, Zoetis, Rome, Italy) injected intravenously (IV) via the external jugular vein and an elastic adhesive bandage was applied to cover the eyes. The computed tomography (CT) images were acquired before and after the manual injection of iodinate contrast medium (600 mgI/kg) (Iopamigita^®^, Insight Agent GmbHR, Heidelberg, Germany) [6] IV via the external jugular vein through a 22 gauge or 18 gauge intravenous catheter. 

Between October 2016 and December 2017, sea turtles with clinical and/or radiographic signs of pulmonary pathologies underwent transtracheal wash, using a 3.3 × 50 mm sterile buster canine urinary catheter and sterile saline solution (NaCl 0.9%). Between January 2018 and June 2019, BAL analysis was performed with endoscopy video system, equipped with a camera and fibro-bronchoscope (Tele Pack Vet x Led^®^ video system, equipped with a Telecam^®^ camera and Karl Storz fibro-bronchoscope) with an operating length of 85 cm and 5.2 mm diameter at the tip. The additional technical characteristics of the fiberscope were: total length of 113 cm, ventral dorsal angular deflection of 195°/105°, visual direction 0°, visual opening angle of 110°, and internal diameter of the working channel equal to 2.3 mm. 

BAL was performed under general anesthesia through the administration of 5–8 mg/kg intravenous propofol (Propovet^®^, Zoetis, Rome, Italy) via the external jugular vein. Before each endoscopic examination, the fiberscope was placed for 10 min in a solution of Tetracycethylenediamine 2% and demineralized water. In 4 out of 14 sea turtles a coelioscopy examination via the prefemoral fossa was also performed. During endoscopic examination, the loggerhead sea turtles were placed in ventral recumbency on a foam mattress in a reverse Trendelenburg position. To reduce turtle stress, their eyes were blinded by applying a self-adhesive elastic bandage around the head. Canine mouth gag retractors were used to keep open the rhamphotheca. To reduce glottis resistance and maintain the sterility of the endoscope, a sterile vaginal speculum for small animals was gently introduced between the branches of the glottis. When the pulmonary disease was presumed to be unilateral, by means of clinical and radiographic findings, the suspected uninvolved lung was investigated first. 

Rotation of the endoscope by 30–45° either direction allowed passage through the bronchus into the lung. BAL was always performed after endoscopic procedures. Sterile 0.9% (NaCl) saline solution (2–3 mL/kg) was applied through a sterile syringe via the endoscope’s irrigation channel and immediately re-collected. The resulting sample was analyzed cytologically and microbiologically. 

In total, 14 BAL samples were collected for bacteriological investigation and susceptibility testing. The samples were cultured on Columbia blood agar (CBA), Mannitol Salt Agar (MSA) and McConkey agar (MCK) (Liofilchem, Teramo, Italy), and incubated in aerobic conditions at 35 °C for 48 h. Bacteria were biochemically identified by means of API-system (Biomérieux, Marcilly-le-Châtel, France). The isolates were tested for susceptibility to 13 antibiotics using the Kirby–Bauer method on Mueller–Hinton Agar (MH, Liofilchem), and the EUCAST epidemiological cut-offs for each antibiotic were used to determine the susceptibility of the tested isolates [7]. The following antibiotics were tested (Liofilchem), (disk abbreviation code and concentration in brackets): ampicillin (AMP; 10 μg), amoxicillin + clavulanic acid (AMC; 30 μg), cephalexin (CFX; 30 μg), ceftazidime (CAZ; 30 μg), cefuroxime (CXM; 30 μg), imipenem (IMI; 10 μg), gentamicin (CN; 30 μg), enrofloxacin (ENR; 15 μg), norfloxacin (NOR; 15 μg), ciprofloxacin (CIP; 5 μg), tetracycline (TET; 30 μg), Trimethoprim-sulphamethoxazole (SXT; 1.25/23.75 μg), lyncomicin (LY; 10 μg) and colistin sulfate (CS; 30 μg) (Liofilchem). After disk applications, the plates were incubated at 37 °C for 24 h. The measurement of zone diameters was interpreted, and strains were categorized as susceptible (S), intermediate (I) or resistant (R). *Escherichia coli* strain ATCC 25,922 was used as quality control.

When deemed necessary, pulmonary physiotherapy was performed via the unilateral or bilateral coupage technique, in order to stimulate the removal of the secretions from the smallest and deepest airways. 

After the hospitalization, sea turtles were discharged to the rescue centers before being released into the sea. 

### Ethical Statement

All sampling procedures were carried out by veterinary personnel. The sampling of turtles from the Department of Veterinary Medicine was conducted with permission of the Department of Veterinary Medicine Animal Ethic Committee (Authorization n.4/19). The involved procedures respect the ethical standards in the Helsinki Declaration of 1975, as revised in 2000 and 2008, as well as the applicable national law (D.L. 26/2014). All efforts were made to minimize animal suffering. 

## 3. Results

Data were collected from 14 turtles with clinical and/or radiographic signs of pulmonary disease (Table 1). The curved carapace length (notch to tip) of the sea turtles ranged from 30 to 77 cm (mean 53.5 cm), and CCW ranged from 28 to 69 cm (mean 48.5 cm). The body weight ranged from three to 51.7 kg (mean 21.1 kg). On clinical evaluation, 11 out of 14 loggerhead sea turtles exhibited sensory depression, poor nutrition and moderate dehydration; 8 turtles showed buoyancy abnormalities, 4 symmetrical and 4 asymmetrical (3 upward left side and 1 right side), and/or inability to submerge; 4 turtles presented with entanglement of flipper or neck lesions, while 1 subject had a longitudinal traumatic lesion at the level of the first three left costal scutes of the carapace. Body temperatures ranged from 18 °C to 22 °C. 

Upon radiographic examination, four of the turtles exhibited severe pneumocoelom. Radiographic signs of pulmonary pathology were seen in all cases; 4 unilateral and 10 bilateral. In these turtles, an increased pulmonary radiopacity, with areas of inhomogeneous parenchyma, interstitial pattern and/or alteration of the pulmonary profile, was found (Figure 1). In one turtle, the radiographic examination showed a reduction of the right lung volume.

Four sea turtles underwent advanced diagnostic investigations with MDCT examination to better characterize the pulmonary pattern. This allowed the investigators to carry out a more precise identification of the areas of pulmonary consolidation. Moderate to severe pneumonia, pneumocoelom and pulmonary reticular pattern were found in all sea turtles.

In three sea turtles, the pneumogastric and pneumohepatic ligaments were stretched due to severe pneumocoelom that crushed the lung parenchyma against the carapace (Figure 2). In two turtles, the dorso-lateral displacement of the trachea and main bronchus was observed.

To assess the resolution of the disease or the improvement of the health of the turtles, imaging (radiography and three cases of MDCT) follow-up was performed after three months.

Blind BAL was performed in eight turtles; the remaining six turtles underwent bronchoscopic-guided BAL.

During endoscopic examination, one sea turtle showed a tracheal involvement, with a hyperemic and congested mucosa and dense exudate on the ventral surface. In two sea turtles, unilateral or bilateral dense and tenacious exudate at the bronchial bifurcation was found. Caliper reduction of the principal bronchial diameter was found in one sea turtle with abundant muco-purulent exudate. In four sea turtles, an intrapulmonary bronchial involvement of the mucosa was seen, which appeared edematous and congested (Figure 3). Only one case showed a slight deviation in the dorso-medial direction of the bronchial path of the left lung, with evident morphological alteration of the faveoli, which appeared oval and crushed transversely in a ventro-dorsal direction. Out of the 14 sea turtles with evidence of pneumocoelom, 4 underwent a coelioscopy procedure. The ventral surface of the lungs was easily identified during coelioscopy, and positive pressure ventilation highlighted lung trauma and laceration in all four turtles.

Cytological examination of bronchial washes was performed in all 14 sea turtles. In total, 12 samples revealed pathological findings: in 2, an aseptic/non-specific heterophilic inflammation was observed, whereas in 10 specimens a moderate to severe inflammation, with toxic heterophils and polymicrobial population evidenced by phagocytic activity of the macrophages, was detected (Figure 4). The presence of fungal hyphae was observed in one sample. Culture and susceptibility tests were performed on respiratory swab samples of all the 14 loggerhead sea turtles. One of these was found to have no bacterial growth. In the other 13 swabs, Gram-negative organisms were found and identified as follows: *Vibrio algynoliticus* (1/13), *Vibrio parahaemolyticus* (1/13), *Pseudomonas aeruginosa* (1/13), *Pseudomonas putrefaciens* (1/13), *Citrobacter freundii* (4/13), *Klebsiella oxytoca* (1/13) and *Aeromonas hydrophila* (4/13). In one swab, a fungal colony was detected and identified as *Candida* spp.

The isolates exhibited resistance percentages higher than 70% against all the beta-lactam molecules tested. In particular, 100% of the strains tested as resistant against ampicillin and imipenem, and 92% tested as resistant against amoxicillin + clavulanic acid, cephalexin and ceftazidime. Seventy percent of the strains were found to be resistant against cefuroxime. Moreover, 100% resistance was also reported against tetracycline and colistin sulfate, as shown in Scheme 1. In addition, two isolates, *Citrobacter freundii* and *Pseudomonas putrefaciens*, were found to be resistant to all the molecules tested.

A specific antibiotic therapy was formulated for seven sea turtles based on enrofloxacin (5 mg/kg, IM, SID), and for four sea turtles based on ceftazidime (20 mg/kg, IM, EOD), in both cases up to 15 days. In two turtles, due to the resistance against all the molecules tested, the antibiotic therapy was not carried out. Regarding the animal with mycotic infection, itraconazole was used for 30 days (5 mg/kg SID per OS). Due to evidence of severe inflammation in the BAL cytology, three sea turtles were administered steroidal anti-inflammatory drug SAID (dexamethasone) at 0.1 mg/kg IM for 15 days BID, but in the case of improvement in the last 5 days, the drug was administered SID. In all cases, tramadol (5 mg/kg, IM/IV, SID) was used as analgesic support. In addition to drug therapy, pulmonary physiotherapy (coupage) was performed in four subjects. A reduction of pneumocoelom was performed by coelomocentesis from the left inguinal fossa in four sea turtles. The turtles mentioned in our study underwent a hospitalization period of between 6 and 149 days (on average 49 days). All the sea turtle underwent radiographic follow ups. The signs of pneumonia resolved over a period of 28 to 70 days. In the two turtles that were subjected only to physiotherapy, the radiographic signs of improvement were obtained over a longer period (80–130 days).

## 4. Discussion

As previously reported, pulmonary disease is common in sea turtles. Due to their unique pulmonary anatomy, carapacial trauma can be a cause of deep lung lesions in sea turtles. Previous studies report that secondary bacterial or other opportunistic infections often occur secondary to deep trauma [2] or opportunistic infections. Furthermore, severe debilitation, i.e., serious illness, depression and other diseases, can prevent normal respiratory acts, causing drowning, as well as forced immersion or broncho-inspiration of water [8]. Inhalation of water is considered one of the primary causes of pneumonia in sea turtles, and secondary bacterial and fungal pulmonary infection is often seen following aspiration [3,4].

The peculiar anatomy of the lower respiratory tract is of clinical importance in chelonians. Inflammatory exudates, particularly those associated with infectious diseases, tend to accumulate in the portion of the lung influenced by gravity (dependent portions). This location precludes timely elimination through the bronchi and trachea, as one would expect in mammalian patients [9,10].

As evidenced by the results of the study, 57% (8/14) of the sea turtles had symptoms of respiratory disease, while the remaining 43% (6/14) of the animals were brought to the STC for different reasons. Routine radiographic examination revealed a pathological pulmonary pattern in all 14 sea turtles examined. These findings confirmed that sea turtles, unlike mammals, do not typically exhibit respiratory disease through dyspnea, cyanosis of the mucous membranes and nasal discharge, but potentially more commonly through buoyancy, which can be detected symmetrically or asymmetrically. Sea turtles use their large lung capacities to regulate buoyancy. During apnea, lung volume determines either a positive buoyancy (float), a negative buoyancy (sink) or a neutral buoyancy (neither float nor sink). Buoyancy disorders are classified as positive, floating or negative, and turtles that cannot achieve positive buoyancy have difficulty getting to the surface of the water to breathe. The main causes of buoyancy are attributable to gastrointestinal floaters, pneumocoelomic floaters, pneumonia, neurological trauma or other diseases [11]. For these reasons, our results suggest that the systematic association of clinical and radiographic examinations can enhance the diagnostic procedures. Radiographic examination is considered the gold standard for screening sea turtles that show respiratory signs or abnormal buoyancy. Our experience confirmed that the slow respiratory rate of reptiles, associated with the lung–air contrast, allowed the lungs to be visualized in detail, suggesting that this might be a technique suitable for diagnosing a pulmonary disease. Out of 14 turtles evaluated, 4 had pneumocoelom.

MDCT enabled us to study the pulmonary parenchyma in depth, which helped to visualize large anatomical segments with greater spatial resolution, and study much more in detail the consolidation of the lungs. Most significantly, MDCT allowed us to study details of the pathological process of pneumocoelom in four turtles that were not well shown on the radiographic examination, and to evaluate the characteristics of the pulmonary pattern. CT provides excellent detail of the internal structures of chelonians, and is the preferred imaging method for the respiratory system in comparison to radiographic examination, and for planning a correct therapeutic approach.

The use of propofol^®^, ranging from 5 to 8 mg/kg, allowed a sufficient degree of anesthesia for the procedures, which had a maximum run of 10–15 min. Moreover, the use of the Total Intravenous Anesthesia (TIVA) protocol during bronchoscopy allowed the avoidance of intubation, due to the high capacity of these animals to maintains apnea for a long period [12]. This provided the greater ease in endoscopic maneuvers.

Of the 14 sea turtles examined, 6 (43%) were subjected to an endoscopic investigation that was a simple and rapid procedure [13]. Hyperemia and edema of the mucosa with or without exudate have been highlighted, especially in the right broncho-pulmonary passage, due to the anatomical location of the right bronchus being more ventral then the left. In four cases, a celioscopy was performed to investigate the presence and cause of pneumocoelom, and in all these cases it was reduced by aspiration from the left inguinal fossa. The presence of pneumocoelom in turtles with pulmonary disease can be considered a consequence of the anatomy of these animals. Lung tears are one of the most frequent causes of pneumocoelom in sea turtles, and turtles with gas within the coelomic cavity, outside of the GI tract and lungs, are classified as pneumocoelomic floaters [5]. Testudines lack a closed pleural cavity as seen in mammals, resulting in a large amount of air lost to the celomic cavity, following lung rupture or pneumonia. The peculiar airway arrangement is potentially detrimental in the case of small lung rupture or perforations, because a direct communication between the main bronchus and the peripheral rupture via the secondary bronchus will result in a large air passage into the coelomic cavity [14]. Coeliomiocentesis was performed in the four subjects with total resolution of buoyancy, and we suppose that buoyancy was an event secondary to lung rupture or pneumonia.

In 8 sea turtles out of 14, BAL analysis was conducted blindly with a buster catheter, and in the remaining 6 sea turtles a bronchoscope BAL was performed. Regardless of the technique used, it was possible to carry out both cytological and microbiological diagnosis. In the bronchoscope wash, an instillation of 2–3 mL/kg of sterile saline solution was found to be sufficient, clearly lower than the values reported in the literature [5]. Thus, in the blind BAL procedure the amount of saline solution administered should be higher compared to the 5 mL/kg described in the literature [5]. BAL procedures, performed transtracheally or endoscopically, made it possible to obtain a diagnosis. However, the use of endoscopy allows one to obtain a more accurate diagnosis of the deep airways, without the risk of contamination of the upper airways. Furthermore, endoscopic washing allows the use of lower saline volumes compared to blind washing, as it is more targeted on the pathological site.

Microbiological examination of the washing liquid showed the presence of bacteria in 93% of the cases, as only 1 out of 14 was found to be bacteriologically negative. Consistent with the literature [15,16,17,18,19,20,21,22,23], all the isolates found in this study were identified as Gram-negative bacteria. These organisms are regarded as opportunistic pathogens, becoming aggressive when other source of stress (i.e., poor water quality, trauma etc.) suppress the turtle’s normal immune response [4,24,25]. In addition, many of these microorganisms might represent a health risk for other marine animals and humans sharing the same environment [26,27], not only because of the pathogenic potential, but also because of the possible dissemination of AMR (antimicrobial resistance) determinants [28,29]. Many studies performed on fish and marine mammals strongly support the hypothesis that wild marine species harbor AMR microbial species, and therefore may serve as a reservoir for AMR genes [28,30]. Gram-positive bacteria were not detected, although their presence, albeit rare, is reported in the literature [22]. Culture-based studies of sea turtle bacterial flora have assessed the antimicrobial susceptibility of bacterial isolates [31,32], and clinically relevant antimicrobial resistance has been noted in a variety of bacteria isolated from sea turtles during antibiotic treatment in rehabilitation centers [22]. On the contrary, here we describe the isolation of resistant bacteria from wild animals that have never been subjected to antibiotic therapy. Nevertheless, in the susceptibility test, 100% of the tested strains were found to be resistant to ampicillin, imipenem, colistin sulfate and tetracycline. Between 70% and 92% resistance was recorded for amoxicillin + clavulanic acid, cephalexin, ceftazidime and trimethoprim-sulphamethoxazole. A high percentage of resistance against beta-lactams antibiotics among marine species has been reported, such as in sea lions (*Zolaphus californianus*), harbor seals (*Phoca vitulina*) and elephant seals (*Mirounga augustirostris*) [33], and focusing on the Mediterranean zone, several reports described high percentages of Gram-negative isolates from both urban river [34] and marine waste water [35]. Conversely, no report has described a similar percentage of resistance against imipenem (100%), which is a broadly active antibiotic that is actually still available for systemic use in human medicine, against mixed multi-drug bacterial infections including opportunistic bacteria such as *Pseudomonas* spp. Therefore, the high percentage of resistance against imipenem in these isolates, discovered in free-ranging loggerheads, should also be investigated, because a previous report in the same Mediterranean loggerhead sub-population described zero resistance against carbapenems [28]. In addition, carbapenemases-producer isolates are often co-resistant to non-beta-lactams molecules such as sulphonamides, tetracyclines, aminoglicosydes and quinolones [36]. Indeed, we reported high resistance rates against Sulphonamides (70%), Tetracyclines (100%), Aminoglicosydes (54%) and fluoroquinolones (lower than 50%), as reported before in the loggerheads from the Mediterranean zone [25,28].

As reported in the literature [5], erythrocytes and bacteria are considered pathological findings, and numerous histiocytes and heterophils suggest an inflammatory infectious process even in the absence of bacteria or fungi. In 85% (12/14) of the cytological samples, the atypical finding of an inflammatory population of heterophil cariolitic granulocytes in phagocytosis, cocciform and bacilli-like bacterial microorganisms, scattered on the bottom, was compatible with a polymicrobial septic heterophilic inflammation, sometimes suppurative or with aspects of chronicity.

The study has therefore highlighted the usefulness of performing the microbiological analysis together with the cytological one, in order to reach a diagnosis as accurate as possible in the shortest time. Cytology is a valid resource in the treatment of lung disease and in the identification of atypical cells, including etiological agents. The antibiogram, performed following bacterial isolation, allows one to undertake the correct therapeutic procedure for the benefit of the patient, avoiding antibiotic-resistance phenomena. This may appear surprising, but sea turtles have been shown to have an astonishing capability to heal, given proper supportive care and husbandry [37]. Before using antibiotics in these animals, it should probably be considered whether they can improve with proper management and physiotherapy alone.

In addition to target antibiotic therapies, the use of pulmonary physiotherapy, through the coupage technique, was useful. In two turtles that had bacterial infection-sustained antibiotic-resistant bacteria, the coupage, together with correct environmental management associated with adequate nutrition, allowed the resolution of the lung disease without antibiotics. In these two cases, control radiographic findings showed the slower but complete resolution of interstitial pneumonia signs.

This study has some limitations, related mainly to the small number of turtles examined. As bronchoscopies require specific skills and are costly, we evaluated a simpler method of obtaining BAL fluid, that is, by a catheter introduced blindly into the bronchial tree of the turtles, and in these cases contamination of culture may be more likely. However, the cytology of the blind BAL allowed us to verify that the sample obtained was derived from the deep airways, and that the presence of high numbers of heterophils suggests an inflammatory infectious process, not only a contamination.

## 5. Conclusions

The early diagnosis of pulmonary disease is a key feature of sea turtle medicine. As highlighted in our study, not all subjects showed respiratory symptoms (i.e., buoyancy). Our research suggests that it may be useful to implement diagnostic procedures, including clinical examination, radiographic and endoscopic examination, and cytological and microbiological testing in order to obtain an accurate diagnosis, to prevent unnecessary therapy and to contain antibiotic resistance phenomena.

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
