# Peer review of "Diagnosis and Treatment of Pulmonary Disease in Sea Turtles (Caretta caretta)"

_animals, 2020, doi:10.3390/ani10081355_

Round 1

Reviewer 1 Report

The manuscript on ‘Diagnosis and Treatment of Pulmonary Disease in Sea Turtles’ written by Ciccarelli et al. provides some useful information regarding the diagnosis and treatment of respiratory pathologies seen in loggerhead sea turtle. While some of the data seem new, the manuscript in general is very poorly structured. Especially the discussion does resemble a collection of short descriptive sentences about the different topics presented, without any profound discussion of the available literature and the implications of the authors findings.

For example, in lines 275-281 the authors discuss their assessment of the radiographic examinations, concluding that ‘this might be a technique suitable for diagnosing a pulmonary disease’, which is hardly a surprise regarding the use of radiographic imaging to detect respiratory diseases in other vertebrates.

Then, in lines 282-287 the authors discuss some questions regarding the anesthesia protocol used, a topic without a connection with the preceding or the following paragraphs.

In lines 288-291 the authors discuss their MDCT results, concluding that ‘MDCT allowed to study details of the pathological process that were not well shown on the radiographic examination and to evaluate the characteristics of the pulmonary pattern’, but without detailing which diagnosis was only possible using MDCT when compared to x-ray imaging.

Then, in lines 292-299 the authors start discussing in only one sentence their endoscopic intrapulmonary examinations, just to follow up with their celioscopy examinations. This does not make sense. The entire part given in lines 295-299 does not make a lot of sense to me. How does ‘The presence of pneumocoeloma in turtles with pulmonary disease can be considered a consequence of the anatomy of these animals.’?  Why is the airway arrangement potentially detrimental in the case of small lung rupture or perforations? If the authors would like to point out that Testudines lack a closed pleural cavity as seen in mammals, resulting in a large amount of air lost to the celomic cavity due to lung rupture, they should explain their reasoning in more detail.

The next part (lines 300-309) the authors jump to a discussion on the possible sampling techniques to perform lung lavage in turtles. If such a comparison was an objective of the study, the authors should state so in the methods section, or they could discuss the importance and efficiency of the different sampling methods in greater detail.

Regarding the antibiotic resistance seen within the cultured bacteria the authors fail to discuss their results with any study about antibiotic resistance. Are the results seen here typical for the antibiotica used and bacteria found? Within reptiles? Within mammals? These results need to be discussed in a much more profound manner.

Why does the last paragraph of the discussion give the same information as the last paragraph of the results section?

The authors emphasize in their conclusion that the floating behavior of turtles is often the only symptom to identify respiratory pathologies in sea turtles and present their data in Table 1. However, they do not present any discussion about the normal buoyancy control in turtles (buoyancy control has been investigated in turtles since the 1970ies) and how the different pathologies they found affected floating in the individual animals. The authors could provide much more details regarding the swimming behavior of turtles with respiratory system pathologies to help others to diagnose such pathologies.

Some minor issues I have identified are:

Line 3: Please include Caretta caretta into the title.

Line 71: ‘prolonged lack of oxygen’ is of course lethal to turtles. A large anoxia tolerance has been found in some species of chelonians, but not in sea turtles that possess greater metabolic demands than other chelonians.

Line 97: The materials and methods section should be much better structured. Where did the animals come from? How were they maintained? Which experimental procedures were undertaken? A better structure avoids unnecessary repetitions.

Line 101: ‘bycatch’ instead of ‘bycaught’

Line 111: The ‘standard (pulmonary) acquisition algorithm’ has been tested for chelonian lungs or is perfected to be used in structurally different mammalian lungs?

Line 116: Why was iodinate contrast medium injected? Do you present any data from these injections, such as pulmonary wasculature

Line 127: ‘Through’ instead of ‘thorough’

Line 154:  Why are the data in Graph 1’ presented in the methods section? Are these not results from the current study? If so, they should be adequately presented in the results section and not in the methods section.

Line 166: Please give the data regarding body temperature you measured.

Line 185: The x-ray given in Figure 1 is difficult to interpret without any comparison with a healthy lung. The dorsal view is not informative, since the viscera ventrally to the lungs reduce the clarity of the x-ray image. What does suggest pulmonary disease in the images presented? The reticular pattern? This is typical of faveolar parenchyma seen in turtle lungs. The accentuation of the intrapulmonary bronchi? It is known that turtle lungs possess more respiratory epithelium in the anterior part of the lungs. Therefore, please provide more convincing data regarding the pulmonary disease identifiable by x-ray images.

Line 196: Figure 2 is not informative at all. What does the reader see in the given image? Where are the lungs? The arrows are lacking. MDCT images could be presented in a much better way than just one single image. The authors have experience with presenting good quality CT-scans (Ricciardi, M.; Franchini, D.; Valastro, C.; Ciccarelli, S.; Caprio, F.; Assad, A.E.; Di Bello, A. et al. 409 Multidetector computed tomographic anatomy of the lungs in the Loggerhead Sea turtle (Caretta caretta). 410 Anat. Rec. 2019, 302, 1658-1665, doi: 10.1002/ar.24030. 411), and should do so here as well.

Line 217: Where does one see tissue ‘that appeared edematous and congested, morphological alteration of the faveoli’. The image presented does not highlight any of these alterations.

Author Response

Response to Reviewer 1 Comments

Thank you for your helpful comments. We have revised our paper accordingly and feel that your comments helped clarify and improve our paper.

Please find our response (in blue) to reviewer’s specific comments (in black) below.

The manuscript on ‘Diagnosis and Treatment of Pulmonary Disease in Sea Turtles’ written by Ciccarelli et al. provides some useful information regarding the diagnosis and treatment of respiratory pathologies seen in loggerhead sea turtle. While some of the data seem new, the manuscript in general is very poorly structured. Especially the discussion does resemble a collection of short descriptive sentences about the different topics presented, without any profound discussion of the available literature and the implications of the authors findings.

For example, in lines 275-281 the authors discuss their assessment of the radiographic examinations, concluding that ‘this might be a technique suitable for diagnosing a pulmonary disease’, which is hardly a surprise regarding the use of radiographic imaging to detect respiratory diseases in other vertebrates.

Thanks for the note

This statement underline how standard radiology can help identify lung diseases, especially because not all operators that cure sea turtles have advanced diagnostics available in order to get to the diagnosis.

Then, in lines 282-287 the authors discuss some questions regarding the anesthesia protocol used, a topic without a connection with the preceding or the following paragraphs.

Thanks for the note. As suggested, we have moved the anesthesia section to the endoscopy part. In our opinion, it is important to make it clear that short and painless procedures can only be performed with the administration of propofol, without any premedication.

In lines 288-291 the authors discuss their MDCT results, concluding that ‘MDCT allowed to study details of the pathological process that were not well shown on the radiographic examination and to evaluate the characteristics of the pulmonary pattern’, but without detailing which diagnosis was only possible using MDCT when compared to x-ray imaging.

Thanks for the note. As suggested, we added: “MDCT allowed to study details of the pathological process as pneumocoelom in four turtles that were not well shown on the radiographic examination and to evaluate the characteristics of the pulmonary pattern. CT allows excellent detail of the internal structures of chelonians and is the preferred imaging method for the respiratory system in comparison to radiographic examination, and to plan a correct therapeutic approach.”

Then, in lines 292-299 the authors start discussing in only one sentence their endoscopic intrapulmonary examinations, just to follow up with their celioscopy examinations. This does not make sense. The entire part given in lines 295-299 does not make a lot of sense to me. How does ‘The presence of pneumocoeloma in turtles with pulmonary disease can be considered a consequence of the anatomy of these animals.’?  Why is the airway arrangement potentially detrimental in the case of small lung rupture or perforations? If the authors would like to point out that Testudines lack a closed pleural cavity as seen in mammals, resulting in a large amount of air lost to the celomic cavity due to lung rupture, they should explain their reasoning in more detail.

Thanks for the note. As suggested, we added “The presence of pneumocoelom in turtles with pulmonary disease can be considered a consequence of the anatomy of these animals. When excess gas is present outside of the GI tract and lungs within the coelomic cavity sea turtle are classified as pneumocoelomic floaters. Pneumocoelom, most commonly caused by lung tears is one of the most frequently diagnosed. Testudines lack a closed pleural cavity as seen in mammals, resulting in a large amount of air lost to the celomic cavity following to lung rupture or pneumonia. The peculiar airway arrangement is potentially detrimental in the case of small lung rupture or perforations because a direct communication between the main bronchus and the peripheral rupture via secondary bronchus will result in a large air passage into the coelomic cavity [13]. Coeliomiocentesis was performed in the 4 subjects with total resolution of buoyancy and we suppose that buoyancy was an event secondary to lung rupture or pneumonia.”

The next part (lines 300-309) the authors jump to a discussion on the possible sampling techniques to perform lung lavage in turtles. If such a comparison was an objective of the study, the authors should state so in the methods section, or they could discuss the importance and efficiency of the different sampling methods in greater detail.

Thanks for the note. As suggested, we added: “BAL procedures, performed transtracheally or endoscopically, made it possible to obtain a diagnosis. However, the use of endoscopy allows to obtain a more accurate diagnosis of the deep airways, without the risk of contamination of the upper airways”

Regarding the antibiotic resistance seen within the cultured bacteria the authors fail to discuss their results with any study about antibiotic resistance. Are the results seen here typical for the antibiotica used and bacteria found? Within reptiles? Within mammals? These results need to be discussed in a much more profound manner.

Thanks for the note. As suggested The microbiological examination of the washing liquid showed the presence of bacteria in 93% of the cases, as only one out of 14 was found to be bacteriologically negative. Consistent to the literature [14,15,16,17,18,19,20,21,22], all the isolates found in this study were identified as Gram-negative bacteria. These organisms are regarded as opportunistic pathogens, becoming aggressive when other source of stress (i.e. poor water quality, trauma ecc…) suppress the turtle’s normal immune response (Innis et al., 2009; Vega-Manriquez et al., 2018; Pace et al., 2019). In addition, many of these microorganisms might represent a health risk for other marine animals and humans sharing the same environment (Santoro et al., 2008; Warwich et al., 2013), not only because of the pathogenic potential, but also because of the possible dissemination of AMR determinants (Foti et al., 2009; Al-Bahry et al., 2011). Many researches performed on fish and marine mammals strongly support the hypothesis that wild marine species harbour  AMR (antimicrobial resistance) microbial species and therefore may serve as reservoir for AMR genes (Miranda and Zemelman, 2001; Foti et al., 2009). Gram-positive bacteria were not detected, although their presence, albeit rare, is reported in literature [22]. Culture-based studies of sea turtle bacterial flora have assessed the antimicrobial susceptibility of bacterial isolates (Harms et al., 2006; Al-Bahry et al., 2009) and clinically relevant antimicrobial resistance has been noted in a variety of bacteria isolated from sea turtles during the antibiotic treatment in rehabilitation centers (Innis et al., 2014). On the contrary, here we describe the isolation of resistant bacteria from wild animals that have never been subjected to antibiotic therapy. Neverthless, in the susceptibility test, 100% of the tested strains were found to be resistant to ampicillin, imipenem, colistin sulfate and tetracycline. From 70 to 92% of resistance was recorded for amoxicillin + clavulanic acid, cephalexin, ceftazidime and trimethoprim-sulphamethoxazole. A high percentage of resistance against beta-lactams antibiotics among marine species have been reported, such as in sea lions (Zolaphus californianus), harbor seals (Phoca vitulina), elephant seals (Mirounga augustirostris) (Frazzon, 2017), and focusing on Mediterranean zone, several reports described high percentages in Gram-negative isolates from both urban river (Maravić et al., 2016) and marine waste water (Maravić et al., 2015). Conversely, no report described a similar percentage of resistant against imipenem (100%), which is a broadly active antibiotic actually reported as still available for systemic use in human medicine, against mixed multi-drug bacterial infections including opportunistic bacteria such as Pseudomonas spp.. Therefore the high percentage of resistance against imipenem in these isolates, discovered in free-ranging loggerheads, should be investigated also because a previous report in the same Mediterranean loggerhead sub-population, described no resistance against carbapenems (Foti et al., 2009). In addition, carbapenemases producers isolates, are often co-resistant to non-beta-lactams molecules such as sulphonamides, tetracyclines, aminoglicosydes, and quinolones (Dandachi et al., 2018). Indeed, we reported high resistance rates against Sulphonamides (70%), Tetracyclines (100%), Aminoglicosydes (54%), and fluoroquinolones (lowerer than 50%) as reported before in the loggerheads from the Mediterranean zone (Foti et al., 2009; Pace et al., 2019).

Why does the last paragraph of the discussion give the same information as the last paragraph of the results section?

Thanks for the note. We have deleted this information

The authors emphasize in their conclusion that the floating behavior of turtles is often the only symptom to identify respiratory pathologies in sea turtles and present their data in Table 1. However, they do not present any discussion about the normal buoyancy control in turtles (buoyancy control has been investigated in turtles since the 1970ies) and how the different pathologies they found affected floating in the individual animals. The authors could provide much more details regarding the swimming behavior of turtles with respiratory system pathologies to help others to diagnose such pathologies.

Thanks. We added: Sea turtles utilize their large lung capacity to regulate their buoyancy. The lung volume during a breath-hold determines whether the turtle will be positively buoyant (float), negatively buoyant (sink), or neutrally buoyant (neither float nor sink). Buoyancy disorder are classified as positive, floating or negative, in turtles that cannot achieve positive buoyancy have difficulty getting to the surface of the water to breathe. The main causes of buoyancy are attributable to gastrointestinal floaters, pneumocoelomic floaters, pneumonia, neurological trauma or other diseases [23].

Some minor issues I have identified are:

Line 3: Please include Caretta caretta into the title.

We added Caretta caretta into the title

Line 71: ‘prolonged lack of oxygen’ is of course lethal to turtles. A large anoxia tolerance has been found in some species of chelonians, but not in sea turtles that possess greater metabolic demands than other chelonians.

Thank for the note. We have modified: “Over the last few decades, some authors have reached the awareness that the inhalation of water can be considered as one of the primary causes of bacterial and fungal pneumonia in sea turtles. Moreover, any debilitating condition can lead to secondary bacterial, fungal, viral (i.e fibropapilloma) or parasitic pneumonia [3,4]”

However, in the past 4 years we have seen a large number of turtles (over 800) that have been trawled and many of them remained more than 4 hours trapped in underwater nets, without breathing, but survived.

Line 97: The materials and methods section should be much better structured. Where did the animals come from? How were they maintained? Which experimental procedures were undertaken? A better structure avoids unnecessary repetitions.

Thank. We added sentences: Between October 2016 and June 2019, 14 loggerhead sea turtles (Caretta caretta) were referred to the Sea Turtle Clinic (STC) of the Department of Veterinary Medicine of the University of Bari "Aldo Moro", after having been taken to local Adriatic and Ionian Sea turtle rescue centers, for buoyancy issues, carapace lesions and flipper and/or neck entanglements. The animals were found drifting at sea, stranded on the coast or were incidentally captured (bycatch).

Upon admission, physical examination of each turtle was performed, including morphometrics [curve carapace length (CCL) notch to tip, curve carapace width (CCW), and weight] as well as measurement of core body temperature from the cloaca, and blood sampling to measure the plasma biochemical and hematological values. Turtles were kept in tanks with saltwater at 36 ‰ (salinity percentage of the Adriatic Sea) in a room at approximately 25-30 °C until clinical improvement.

We would like to specify that no experimental procedures have been carried out, but only clinical procedures. In our country, these animals are considered “unavailable heritage of the State” and cannot be underwent to experimental.

Line 101: ‘bycatch’ instead of ‘bycaught’

Thank. We changed it

Line 111: The ‘standard (pulmonary) acquisition algorithm’ has been tested for chelonian lungs or is perfected to be used in structurally different mammalian lungs?

Thank for the interesting. As we have described in another article the acquisition algorithm is the one normally used in mammals.

(Ricciardi, M.; Franchini, D.; Valastro, C.; Ciccarelli, S.; Caprio, F.; Assad, A.E.; Di Bello, A. et al. Multidetector computed tomographic anatomy of the lungs in the Loggerhead Sea turtle (Caretta caretta). Anat. Rec. 2019, 302, 1658-1665)

Line 116: Why was iodinate contrast medium injected? Do you present any data from these injections, such as pulmonary wasculature

Thank for the interesting. We have studying inflammation and abscesses. Pathological vascularization is of interest to us in order to go on our study on lungs (see Ricciardi, M.; Franchini, D.; Valastro, C.; Ciccarelli, S.; Caprio, F.; Assad, A.E.; Di Bello, A. et al. Multidetector computed tomographic anatomy of the lungs in the Loggerhead Sea turtle (Caretta caretta). Anat. Rec. 2019, 302, 1658-1665, doi: 10.1002/ar.24030). Unfortunately, to date, we have few data to be able to express certain data.

Line 127: ‘Through’ instead of ‘thorough’

Thank. We changed it

Line 154:  Why are the data in Graph 1’ presented in the methods section? Are these not results from the current study? If so, they should be adequately presented in the results section and not in the methods section.

Thank for the note. We moved the Graph 1 in the results section.

Line 166: Please give the data regarding body temperature you measured.

Thank. We added data body temperature.

It should be noted that the sea turtles were referred to the Sea Turtle Clinic after a variable period of 24-48 hours from the recovery centers.

Line 185: The x-ray given in Figure 1 is difficult to interpret without any comparison with a healthy lung. The dorsal view is not informative, since the viscera ventrally to the lungs reduce the clarity of the x-ray image. What does suggest pulmonary disease in the images presented? The reticular pattern? This is typical of faveolar parenchyma seen in turtle lungs. The accentuation of the intrapulmonary bronchi? It is known that turtle lungs possess more respiratory epithelium in the anterior part of the lungs. Therefore, please provide more convincing data regarding the pulmonary disease identifiable by x-ray images.

Thank for the note.

It is right to consider that DV radiography is not representative for the evaluation of lung fields, however in case of severe lung disease, an alteration of the lungs pattern is still appreciated, despite the overlap with the tissues of the gastrointestinal tract.The evidence of these radiographic aspects testifies to the seriousness of the pathological process

We enclose the example of a DV image of a perfectly healthy turtle, where it is possible to appreciate the pulmonary silhouette, tracheal bifurcation, and vasculature. In the DV image proposed in the article this is not appreciable.

This is an example of DV of healthy sea turtle

The bigger arrows shown pulmonary silhouette, the arrows with A,B and V indicated arterial, bronchus and vein respectively.

Line 196: Figure 2 is not informative at all. What does the reader see in the given image? Where are the lungs? The arrows are lacking. MDCT images could be presented in a much better way than just one single image. The authors have experience with presenting good quality CT-scans (Ricciardi, M.; Franchini, D.; Valastro, C.; Ciccarelli, S.; Caprio, F.; Assad, A.E.; Di Bello, A. et al. 409 Multidetector computed tomographic anatomy of the lungs in the Loggerhead Sea turtle (Caretta caretta). 410 Anat. Rec. 2019, 302, 1658-1665, doi: 10.1002/ar.24030. 411) and should do so here as well.

Thank for the suggested. We have modified the images and the caption; we increase the number of images so that the reader can better understand what has been detected in the CT findings

Line 217: Where does one see tissue ‘that appeared edematous and congested, morphological alteration of the faveoli’. The image presented does not highlight any of these alterations.

we had uploaded B/W images believing that the color figures were paid. Now the reviewer will be able to better appreciate what is described in the captions, with another figure.

Reviewer 2 Report

In general, this information is not novel. There are numerous pulmonary disease papers in sea turtles which you did not cite.

Majority of paper is written in single sentences spaced as if own paragraph. I'd suggest combining into cohesive paragraphs. 

General flow of results and discussion needs major revision - for example have paragraphs on imaging techniques (radiograph and MDCT) follow one another then endoscopy then anesthesia. This will improve flow and reader interest.

Clinical signs such as open mouth breathing, hemoptysis and nasal and oral discharge can be seen in sea turtles with severe pulmonary disease. You make many "factual statements" about pulmonary disease in sea turtles in your results and discussion based off your study, but this is specific to 14 individuals and should not be used to represent all sea turtles.

It is interesting to focus on antibiotic susceptibilities; however you must recognize that contamination of culture may be possible, especially with "blind" BAL (transtracheal wash) which represents 8/14 turtles in study. You will need to mention in the discussion.... it is VERY unlikely that two turtles with resistant pulmonary infections would improve with just coupage. More likely that you grew a contaminant or normal flora from upper respiratory tract. 

You really need to mention and clarify all the limitations of this study - small sample size, majority are traumatic injury and do not represent all types of pulmonary disease, bronchoscope vs. blind BAL, etc.

Author Response

Thank you for your helpful comments. We have revised our paper accordingly and feel that your comments helped clarify and improve our paper.

Please find our response (in blue) to reviewer’s specific comments (in black) below.

In general, this information is not novel. There are numerous pulmonary disease papers in sea turtles which you did not cite.

Majority of paper is written in single sentences spaced as if own paragraph. I'd suggest combining into cohesive paragraphs. 

General flow of results and discussion needs major revision - for example have paragraphs on imaging techniques (radiograph and MDCT) follow one another then endoscopy then anesthesia. This will improve flow and reader interest.

Clinical signs such as open mouth breathing, hemoptysis and nasal and oral discharge can be seen in sea turtles with severe pulmonary disease. You make many "factual statements" about pulmonary disease in sea turtles in your results and discussion based off your study, but this is specific to 14 individuals and should not be used to represent all sea turtles.

It is interesting to focus on antibiotic susceptibilities; however you must recognize that contamination of culture may be possible, especially with "blind" BAL (transtracheal wash) which represents 8/14 turtles in study. You will need to mention in the discussion.... it is VERY unlikely that two turtles with resistant pulmonary infections would improve with just coupage. More likely that you grew a contaminant or normal flora from upper respiratory tract. 

You really need to mention and clarify all the limitations of this study - small sample size, majority are traumatic injury and do not represent all types of pulmonary disease, bronchoscope vs. blind BAL, etc.

Tanks for the suggestion. We corrected all the phrases that had been reported in the pdf. Below are the changes and comments required by the reviewer.

You should mention that any debilitating condition or other causes of poor health can lead to secondary bacterial and/or fungal pneumonia. Also consider other causes of pulmonary disease like virus (fibropapillomas, etc.) or parasites.

Thank for the note. We added: “Over the last few decades, some authors have reached the awareness that the inhalation of water can be considered as one of the primary causes of bacterial and fungal pneumonia in sea turtles. Moreover, any debilitating condition can lead to secondary bacterial, fungal, viral (i.e fibropapilloma) or parasitic pneumonia [3,4].”

You should add in a sentence describing difference in samples obtained from BAL vs. transtracheal wash. You mentioned BAL is sampling lower airways, mention here that samples from tracheal wash are typically only representative of upper airways.

Thank for the comment: we add (upper airways) “Transtracheal wash can be performed "blindly", through a sterile catheter or bronchoscopically, depending on the instrumentation available and the samples obtained are typically representative of upper airways”.

In the literature, this view is typically described as AP (anterior-posterior) for sea turtles

Thank for the comment:

We used the abbreviation CrCd as reported in textbook that describe diagnostics imaging in sea turtles:

-chapter 6 Diagnostic imaging, Sea Turtle Health and Rehabilitation; Manire, C.A.; Norton, T.M.; Stacy, B.A.; Innis, C.J.; Harms, C.A.; Eds; J. Ross Publishing: Plantation, Florida, 2017;

- chapter 29 Diagnostic Imaging, Reptile Medicine and Surgery (2nd Edition) Mader

-chapter 8 Diagnostic imaging techniques, Medicine and Surgery of Tortoises and Turtles. McArthurS, Wilskinson R and Meyer J.

But if the reviewer preferred AP we can change it

This graph is a result. I would move this graph entirely to results where you talk about the isolates and susceptibility testing.

Thank for the comment. We moved the Graph 1 in results

Would be good to include how many isolates were tested. Were all these antibiotics tested for every isolate?

This is an interesting comment.

We added: “The isolates were tested for susceptibility to 13 antibiotics using the Kirby-Bauer method on Mueller-Hinton Agar (MH, Liofilchem) and the EUCAST epidemiological cut-offs for each antibiotic were used to determine the susceptibility of tested isolates [7]. The following antibiotics were tested (Liofilchem), (disk abbreviation code and concentration in brackets): ampicillin (AMP; 10 μg), amoxicillin + clavulanic acid (AMC; 30 μg), cephalexin (CFX; 30 μg), ceftazidime (CAZ; 30 μg), cefuroxime (CXM; 30 μg), imipenem (IMI; 10 μg), gentamicin (CN; 30 μg), enrofloxacin (ENR; 15 μg), norfloxacin (NOR; 15 μg), ciprofloxacin (CIP; 5 μg), tetracycline (TET; 30 μg), Trimethoprim-sulphamethoxazole (SXT; 1.25/23.75 μg), lyncomicin (LY; 10 μg) and colistin sulphate (CS; 30 μg) (Liofilchem). After disks application, the plates were incubated at 37°C for 24 hrs. The measurement of zone diameters was interpreted, and strains were categorized as susceptible (S), intermediate (I) or resistant (R). Escherichia coli strains ATCC 25922 was used as quality control.”

I would mention which isolates were found in these two turtles as this is very interesting.

Thank. This is a right observation. We added: “In addition, two isolates, Citrobacter freundii and Pseudomonas putrefaciens were found to be resistant against all the molecules tested. “

Why?

we apologize for the inconvenience. We reformulated the sentence to make it clear what we meant by half dosage

“In three sea turtles, steroidal anti-inflammatory drugs SAID (dexamethasone) was administered at 0.1 mg/kg IM for fifteen days BID but in case of improvement the last five days, drug was administered SID.”

(As reported in appendix 7 pp988 Sea Turtle Health and Reabilitation Manire, C.A.; Norton, T.M.; Stacy, B.A.; Innis, C.J.; Harms, C.A.

This is very confusing. You never mention doing statistics anywhere in methods. You will need to clarify and describe what tests were used, what p value are you using for significance, etc.

Thanks for the comment. We had forgotten to specify the statistical test used.

We added the statistical test applied: “The chi-square test was applied to compare unilateral or bilateral involvement of the hospitalized turtles (P<0.05). 

You should write that they do not "typically" exhibit respiratory disease through...

.... but "more commonly"

*Sea turtles can present with dyspnea, cyanosis, nasal discharge, hemoptysis, etc. (particularly with severe pneumonias secondary to cold stunning or debilitation rather than trauma). so need to clarify that buoyancy issues were more common in the subjects of your study and this may be due to higher number of traumatic injury in the study

Thank for the commenti. We added “Sea turtles utilize their large lung capacity to regulate their buoyancy. The lung volume during a breath-hold determines whether the turtle will be positively buoyant (float), negatively buoyant (sink), or neutrally buoyant (neither float nor sink). Buoyancy disorder are classified as positive, floating or negative, in turtles that cannot achieve positive buoyancy have difficulty getting to the surface of the water to breathe. The main causes of buoyancy are attributable to gastrointestinal floaters, pneumocoelomic floaters, pneumonia, neurological trauma or other diseases [23].”

There are numerous papers on radiographic pulmonary determination in sea turtle species. None of this is novel.

Last two sentences do not make any sense.

Any thoughts as to why more common on the right side? Describe why here, anatomical location of right bronchus more ventral then left. Information is in the sea turtle medicine textbook that you reference.

Thank for the note. We change this sentence: “Hyperemia, edema of the mucosa with or without exudate have been highlighted, especially on the right broncho-pulmonary passage, due to anatomical location of right bronchus more ventral then left”

This MDCT paragraph should be moved to directly after the radiographic exam paragraph (onto line 282). Keep the endoscopic exam and anesthesia info together.

Thank. We keep the two paragraphs together.

Need to reword this last sentence, it is very confusing

Thank for the note. This is a citation, but we simplify this sentence.

“The peculiar airway arrangement is potentially detrimental in the case of small lung rupture or perforations because a direct communication between the main bronchus and the peripheral rupture via secondary bronchus will result in a large air passage into the coelomic cavity [13].”

Why? You are making a recommendation on volume of saline used in a blind vs. bronchoscopic BAL and need to clarify what you mean here...

Thank for the comment. We added: “BAL procedures, performed transtracheally or endoscopically, made it possible to obtain a diagnosis. However, the use of endoscopy allows to obtain a more accurate diagnosis of the deep airways, without the risk of contamination of the upper airways. Furthermore, endoscopic washing allows the use of lower saline volumes compared to blind washing, as it is more targeted on the pathological site:”

This sentence is confusing. Did you obtain a sample of coupaged fluid???

You need to mention somewhere how significant the radiographic pulmonary changes were on these two turtles. Was there evidence of pneumonia (interstitial or edicular pattern) or potentially just aspiration.

Again, it is critical to remember and state that the results of BAL (particularly "blind" BAL or tracheal wash) may yield a large amount of contaminants or potential "normal" flora of upper respiratory or GIT. Could it be that resistant bacteria on these two cultures was contamination of oral flora and therefore thats why they improved?

Thank for the comment. We added: In addition to target antibiotic therapies, the use of pulmonary physiotherapy, through coupage technique, was useful. In two turtles that had bacterial infection sustained antibiotic-resistant bacteria the coupage, together with a correct environmental management associated with adequate nutrition, allowed the resolution of the lung disease without antibiotics. In these two cases, control radiographic findings found a slower but complete resolution of interstitial pneumonia signs.

Control x-rays were also performed in these subjects. It was not possible to carry out further microbiological analyzes on the expectorate because it was often ingested by the subject or expelled in the tank (therefore contaminated). The bacteria highlighted in these two subjects were Pseudomonas putrefaciens and Citrobacter freundi Furthermore, the cytology associated with the microbiological examination gave us confirmation that the change came from the deepest lung sites (toxic heterophils and polymicrobial population, evidenced by phagocytic activity of the macrophages was detected).

Round 2

Reviewer 1 Report

The manuscript has been significantly improved and I have no further recomendations.

Author Response

As requested by the editor, we have modified the latest clarifications requested by the second reviewer. Thanks for the availability

Reviewer 2 Report

I am sorry, but I have to reject this paper. This information is not novel and does not provide a significant contribution to the scientific literature of sea turtle medicine. These are mainly just observations of pulmonary disease in turtles, which have all been well documented in both papers and textbooks. There is no solid scientific data or statistical analyses and, although the antibiotic resistance portion is interesting and important, it remains flawed by inconsistent and potentially inappropriate sample collection and makes assumptions off of a very small sample size. I do appreciate the acknowledgement of these limitations in the paper, but it is not enough to make this publishable. 

There are also several newly added sentences that are basically plagiarized word from word and not cited appropriately. 

I have attached some revisions on the PDF and I hope this is helpful if you'd like to improve this for the future. 

Author Response

Thank you for your availability. We made the latest changes requested by the editor, changing the plagiarized sentences
